

# Adaptation of RainGaugeQC algorithms for quality control of rain gauge data from professional and non-professional measurement networks

Katarzyna Ośródka, Jan Szturc, Anna Jurczyk, Agnieszka Kurcz

Institute of Meteorology and Water Management – National Research Institute, ul. Podleśna 61, 01-673 Warsaw, Poland

*Correspondence to*: Jan Szturc (jan.szturc@imgw.pl)

**Abstract.** Rain gauge measurements are one of the primary techniques used to estimate a precipitation field, but they require careful quality control. This paper describes a modified RainGaugeQC system, which is applied to real-time quality control of rain gauge measurements made every 10-min. This system works operationally at the national meteorological and hydrological service in Poland. The RainGaugeQC algorithms, which have been significantly modified, are described in detail. The modifications were made primarily to control data from non-professional measurement networks, which may be of lower quality than professional data, especially in the case of private stations. Accordingly, the modifications went in the direction of performing more sophisticated data control, applying weather radar data and taking into account various aspects of data quality, such as consistency analysis of data time series, bias detection, etc. The effectiveness of the modified system was verified based on independent measurement data from manual rain gauges, which are considered one of the most accurate measurement instruments, although they mostly provide daily totals. In addition, an analysis of two case studies is presented. This highlights various issues involved in using non-professional data to generate multi-source estimates of the precipitation field.

## 1 Introduction

### 1.1 Precipitation measurements

Precipitation is one of the most important meteorological parameters – due to its great practical importance in water management, flood control and other issues (e.g., Loritz et al., 2021; Sokol et al., 2021). For this reason, conducting measurements and estimating the precipitation field are very important tasks, though also very challenging because of the very high temporal and spatial variability of precipitation and its intermittent nature. The shorter the accumulation time of measurements, the greater the spatial variability of an estimated precipitation field and the greater its uncertainty (Berndt and Haberlandt, 2018; Bárdossy et al., 2021). This is especially true when estimating sub-daily totals and, even more the case for sub-hourly precipitation totals.

Until today, the basic measurements of precipitation are in situ measurements carried out by means of rain gauge networks, and this does not change despite the intensive development of remote sensing techniques, such as radar and satellite, from which measurements are highly distorted. Rain gauge measurements are still considered the most accurate, although they are limited to specific, rather sparsely distributed points. Consequently, when estimating the precipitation field, measurement data provided by different techniques are treated as independent estimates of the same physical quantity. Thus, the final estimate of a precipitation field, which is often referred to





as quantitative precipitation estimation (QPE), is determined using various methods of combining data from different sources (multi-source estimation), taking into account the strengths and weaknesses of each of these techniques (McKee et al., 2016; Jurczyk et al., 2020b).

Since all measurement techniques are subject to significant errors, which have a different temporal and spatial structure, all rainfall measurements need advanced quality control (QC) (Szturc et al., 2022). This applies not only to weather radar measurements (Ośródka et al., 2014; Ośródka and Szturc, 2022; Sokol et al., 2021), but also to rain gauge measurements. The latter are considered accurate at their locations, however field experiments (Wood et al., 2000) and experiences with dual-sensor rain gauges (Ośródka et al., 2022) show that trust in rain gauges is often excessive – errors in their measurements can sometimes be very significant.

Quality control of rain gauge data is carried out using various approaches, most commonly by analysing the spatial and temporal distribution of measurements. As such information is insufficient for effective QC, especially in the case of sparse measurement networks, external data from other measurement techniques, most often weather radar and satellite, are often used (Ośródka et al., 2022; Yan et al., 2024). Increasingly, deep learning techniques are also being applied for QC (Sha et al., 2021). It should be noted that QC applied to short rainfall totals, such as the 10-min employed in this work, is considerably more difficult than for longer totals, such as 1-h (Villalobos-Herrera et al., 2022).

Due to the particular importance of rain gauge measurements, especially for the adjustment (calibration) of radar and satellite measurements, it is crucial when estimating the precipitation field that rain gauge networks are as dense as possible (Hohmann et al., 2021). This implies a very high financial as well as technical and organisational effort, so that a great deal of work is currently being done to deliver rain gauge data from other networks, not only from the national meteorological and hydrological services (NMHSs). A separate issue is the employment of "opportunistic" measurement techniques, i.e. precipitation data acquired from devices not dedicated to rainfall measurement, e.g. by analysing the attenuation of signals in commercial microwave links used in mobile phone networks, see e.g.: Chwala and Kunstmann (2019), Polz et al. (2020), Graf et al. (2021), Pasierb et al., (2024).

## 1.2 Non-professional rain gauge networks

Apart from the rain gauge networks of the NMHSs, measurement networks set up and maintained by various institutions – usually state or local authorities taking measurements for their own purposes – can also be a source of rain gauge data. Another possibility is collecting meteorological measurements carried out by individual people with generally low-cost measuring stations, for whom taking measurements, analysing them and comparing with data generated by meteorological services is a hobby activity (Muller et al., 2015; Krennert et al., 2018; Zheng et al., 2018). These are so-called private or citizen weather stations (PWS or CWS).

For the purposes of this paper, all measurements carried out by institutions other than NMHSs are considered "non-professional" because they do not guarantee compliance with the standards set by the World Meteorological Organisation (WMO) (WMO-No. 488, 2010) to the same extent as NMHSs measurements. A distinction between professional and non-professional rain gauges has been proposed by, among others, Garcia-Marti et al. (2023). In addition, the aforementioned private stations set up by individual hobbyists need to be distinguished, as direct control of the location, technical conditions or maintenance of such stations is impossible in practice. Such stations should be treated with much less trust, and the high uncertainty of the data is due to a number of reasons, which





have been described in detail in the literature (np. Bell et al., 2015; Båserud et al., 2020; Hahn et al., 2022; Urban
et al., 2024). Nevertheless, many studies show that such data can be a valuable source of precipitation information
(de Vos et al., 2017; 2019; Horita et al., 2018; Nipen et al., 2020; Bárdossy et al., 2021), thanks to the relatively
very large number of these stations especially in urban areas, bearing in mind that professional gauges are typically
located outside city centres (Overeem et al., 2024).
The incorporation of non-professional data is associated with some overall increase in uncertainty in
precipitation data. Moreover, dual-sensor rain gauges are rarely used and this reduces the efficiency of the quality
control performed. Consequently, QC algorithms for these data should include not only the filtering out of clearly
erroneous measurements and a decrease of their quality metric in the form of e.g. a quality index ($QI$), but for less
supervised networks it is also necessary to correct at least the systematic errors associated with the bias of these
measurements.

**1.3 Overview of approaches to QC of rain gauge data**

The specificity of data from non-professional rain gauges is primarily due to the greater uncertainty of their
measurements. This entails the development of more sophisticated, but also more restrictive quality control
algorithms. These are generally extensions of the QC methods applied to data from NMHSs, but here they analyse
the reliability of individual measurements in more depth. These methods most often rely on verification with
professional rain gauges, but also use other measurement data, especially weather radar data.
*Spatial distribution of precipitation measurements* – detection of inconsistencies with surroundings. The most
common quality control techniques involve checking whether the deviation from the reference measurements,
which can also be data from nearby rain gauges, are within preset threshold values. If a measurement exceeds the
threshold, then it is treated as an outlier and either its quality index $QI$ (or quality flag) is decreased or the
measurement is rejected (de Vos et al., 2017; Båserud et al., 2020). In addition, precipitation data from other
sources, primarily weather radar, can be used to quantify the uncertainty of outlying measurement data (Ośródka
and Szturc, 2022). Spatial consistency tests are very difficult to perform for a sparse rain gauge network, so the
QC in terms of spatial consistency may not be carried out, and in the case of private rain gauges, such data may
simply be rejected (Nipen et al., 2020). Alerskans et al. (2022) used a cost function based on a contingency table,
which optimises the parameters of the spatial QC algorithm used to detect as many actually erroneous data as
possible, while minimising the number of correct data that were found to be erroneous.
*Correlation of time series of precipitation measurements with reference data.* The temporal consistency check
involves detecting stations from which measurements often have relatively low reliability, but not so much that
individual measurements do not pass a spatial consistency check. Analysis of the temporal consistency of rainfall
data is most often carried out by analysing the correlation of the time series from the controlled rain gauge with
the time series of reference data (Bárdossy et al., 2021; de Vos et al., 2019). Reference data can be either data from
professional rain gauges of relatively high quality or from other measurement techniques, primarily weather radar
(de Vos et al., 2019). However, the use of radar data is associated with difficulties, most often due to errors in
estimation of the precipitation field (Ośródka et al., 2014; Ośródka and Szturc, 2022). Moreover, weather radar
measurements are performed at certain heights above the ground surface – from a few hundred metres to as much
as a few kilometres – and are then spatially averaged. Analysis of the correlation coefficient of a time series
becomes difficult, especially in cases where the rain gauge reports false zero values (no precipitation) due to e.g.


a sensor being blocked or some object obstructing the path of the rain (e.g. buildings, vegetation). Another
difficulty is caused by non-rainfall periods – time series with predominantly very low rainfall can sometimes
disturb the correlation (Hahn et al., 2022).
*Detection and removal of bias in precipitation measurements.* The approaches to the issue of quality control
of rain gauge data described above do not correct erroneously measured values, but only reduce their *QI* or remove
them. However, data correction is an important part of data quality control. First of all, it is about bias correction
(unbiasing), which most often results from rainfall underestimation related to rain gauge technology: rain gauge
measurements are underestimated due to wind-induced errors, wetting losses, evaporation losses, trace
precipitation, etc. The magnitude of the underestimation also depends on the construction of the rain gauge; in
particular, tipping-bucket devices are subject to significant bias (Segovia-Cardozo, 2021). This bias can be
eliminated, or at least reduced, by, for example, quantitatively analysing all underestimation factors and
introducing all important corrections (Zhang et al., 2019). Such adjustments, however, are generally conservative
because of the difficulty of considering all relevant factors and the lack of precise data on influencing parameters.
Another way is to compare non-professional measurements with a benchmark as reliable as possible, which could
be manual rain gauges, preferably lysimetric ones that measure at ground level (Haselow et al., 2019; Schnepper
et al., 2023). However, such measurements are not common. Radar observations are more widely available but
using them as a benchmark requires the QC and adjustment to professional rain gauge measurements to have been
previously carried out. Unbiasing is also calculated on the basis of a larger data set collected during precipitation
events typical of the local climate (np. de Vos et al., 2019). The bias factor determined on this basis is treated as a
climatological quantity.
**1.4 Structure of the paper**
This paper presents the RainGaugeQC system (Ośródka et al., 2022) after its adaptation for quality control of rain
gauge data from non-professional stations. The paper is structured as follows: after Section 1, Section 2 briefly
describes the different kinds of precipitation data on which the RainGaugeQC was developed and verified. Section
3 presents the algorithms of the RainGaugeQC system with the emphasis on solutions that are more advanced
when compared to the earlier version of the system. Results obtained over several months, as well as analysis of
two case studies, were discussed in Section 4. Section 5 summarises the paper with a list of conclusions resulting
from the use of the modified RainGaugeQC system.
**2 Precipitation data**
**2.1 Available networks of rain gauges**
IMGW operationally utilises telemetric rain gauge data from the following measurement networks operated by
(Fig. 1):

• IMGW (Institute of Meteorology and Water Management – National Research Institute) – network of

NMHS in Poland (https://hydro.imgw.pl/#/map).

• CHMU (Czech Hydrometeorological Institute) – network of NMHS in the Czech Republic. IMGW uses

data from more than 324 stations near the Polish border

(https://www.chmi.cz/files/portal/docs/meteo/ok/images/srazkomerne_stanice_en.gif).



• General Directorate of the State Forests (DLP) – network of the meteorological monitoring program of
forest areas consisting of 145 stations (https://www.traxelektronik.pl/pogoda/las/).
The above data are used to generate operationally (in real-time) a multi-source precipitation field with high
spatial resolution, which is the basis for generating nowcasting precipitation forecasts.
Synthetic information about the above networks is summarised in Table 1.

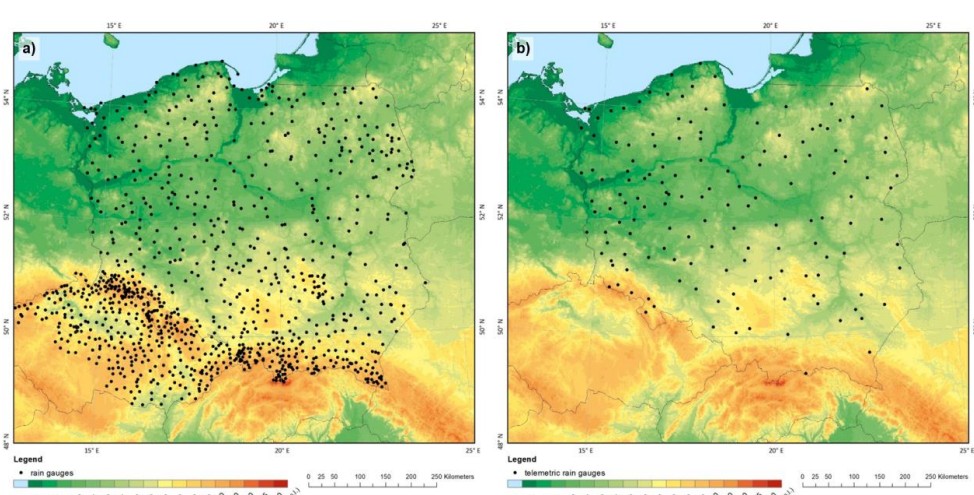

**Figure 1: Telemetric rain gauge networks: a) IMGW and CHMU, b) State Forests.**
**Table 1. Rain gauge networks incorporated into operational processing by RainGaugeQC system and**
**estimation of precipitation field (as of October 2024).**

| ID | Network operator | Number of stations | Type of rain gauges | Type of network |
|----|------------------|--------------------|--------------------|-----------------|
| 1 | IMGW | 656 | Mostly two tipping bucket sensors | Professional |
| 2 | CHMU | 324 stations located close to Polish territory | Mostly tipping bucket sensors | Professional |
| 3 | General Directorate of the State Forests (DLP) | 145 | Heated | Non-professional |


For the domain of Poland data from professional rain gauge networks operated by NMHSs in Poland (IMGW)
and the Czech Republic (CHMU) are available. As the territory of the Czech Republic covers a large part of the
analysed domain and, above all, a significant number of rain gauges are located close to mountainous areas on the
border with Poland (Fig. 1), these data are very important for improving the reliability of the estimation of the
precipitation field in southern Poland. The third network, belonging to the State Forestry Authority, is a non-
professional research network so it is uncertain whether all the standards of the WMO recommendations are
followed (WMO-No. 488, 2010).





The quality of precipitation data is highly dependent on the type of measuring devices being used. Currently,
the IMGW network is still dominated by tipping-bucket type rain gauges, which are considered significantly less
accurate than weighing rain gauges (e.g., Colli et al., 2014; Hoffmann et al., 2016).

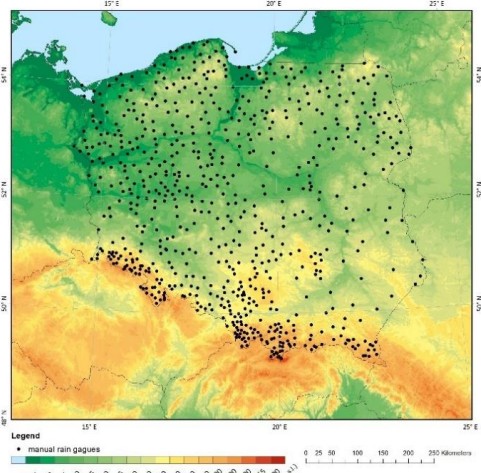

**Figure 2: IMGW's network of manual rain gauges.**

A network of Hellmann-type manual rain gauges, providing independent reference data, is used in this study
to verify the performance of the developed QC algorithms. As the data from these rain gauges are not available in
real time, they cannot be used for rainfall field estimation or operational QC of telemetric data. The IMGW network
consists of about 641 manual rain gauges, which provide daily rainfall accumulations (Fig. 2). These data are
believed to be much more accurate than measurements from telemetric rain gauges, especially tipping-bucket ones.
They are subjected to manual QC before being used.
**2.2 Weather radar precipitation data**
Precipitation data from weather radars play a major role in the RainGaugeQC system for quality control of rain
gauge data (Ośródka et al., 2022). The data used in this study are provided by the Polish POLRAD radar network
operated by IMGW. The network consists of ten Doppler polarimetric radars working in C-band, manufactured by
Leonardo Germany (Fig. 3). Three-dimensional raw data and two-dimensional products are generated by the
Rainbow 5 system every 5 min with 1-km spatial resolution and a range of 250 km.

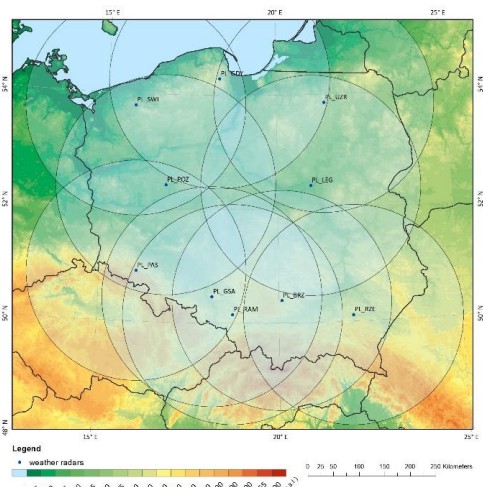


**Figure 3: Computational domain of Poland with plotted 215-km ranges of weather radars of the Polish POLRAD radar**
**network (as of July 2024).**

195  The raw 3D radar data are quality controlled and corrected by the RADVOL-QC system (Ośródka et al., 2014;

Ośródka and Szturc, 2022). The product used to estimate the rainfall field is PseudoSRI (Pseudo Surface Rainfall
Intensity): cut-off at 1-km altitude above ground and from the lowest elevation out of the SRI range, generated
every 5 min and accumulated into 10-min sums taking into account spatio-temporal interpolation between two
adjacent measurements. As a result of quality control with the RADVOL-QC system, the corresponding $QI$ quality
index fields are also assigned to the individual estimated precipitation fields. In addition, some kind of quality
control of radar precipitation takes place at the stage when data from individual radars is combined into composite
maps. This is done by means of an algorithm that takes into account the time-varying spatial distribution of the
quality index (Jurczyk et al., 2020a).

204  Due to the bias present in the weather radar observations, these data are adjusted with rain gauge data, but only

from the professional networks, derived from the 1-h moving window. However, if a precipitation accumulation
is below a preset threshold, then this period is extended accordingly, up to a maximum of the seasonal
accumulation.
**2.3 Multi-source precipitation estimates RainGRS**
Multi-source precipitation field estimates are generated by the RainGRS system of IMGW. The system combines
rain gauge, weather radar, and satellite precipitation data in real time (Szturc et al., 2018; Jurczyk et al., 2020b;
2023). The algorithm for combining these data is based on conditional merging according to an algorithm proposed
by Sinclair and Pegram (2005), which attempts to enhance the strengths and reduce weaknesses of individual
measurement techniques. This approach was modified in RainGRS by taking into account the quantitative
information about the spatial distribution of the quality of the individual input data (quantified by $QI$). These
estimates are produced every 10 min with a high spatial resolution of 1 km x 1 km.





In the study, two versions of multi-source RainGRS precipitation estimates are generated in order to examine
the impact of incorporating non-professional data. In the first version, rain gauge data only from the professional
networks of IMGW and CHMU were taken, while in the second version, data from the non-professional network
of the State Forests were added to this set.
**3 RainGaugeQC system for QC of rain gauge data**
**3.1 RainGaugeQC system for QC of rain gauge data from a professional network**
The RainGaugeQC system was originally designed to perform real-time quality control of rain gauge data from
measurement networks maintained by IMGW. This system was described in detail in work by Ośródka et al.
(2022), so in this study, after a very concise presentation of the algorithms, the following sections will describe
only modifications made to adapt it to data from non-professional networks.
In the standard version of RainGaugeQC (Ośródka et al., 2022) (see column "Before modification" in Table
2), firstly the simple plausibility tests – the gross error check (GEC) and range check (RC) – were performed on
individual measurements. Then the more complex checks were conducted using a larger amount of rain gauge data
from either a specific time range or a specific area, as well as using external data provided by weather radars.
Firstly, the Radar Conformity Check (RCC) was applied to identify false precipitation on the basis of the radar
measurements. Obstruction or blocking of the sensors was also checked for. Next, the Temporal Consistency
Check (TCC) was performed, but this version was designed only for dual-sensor stations: data from the pairs of
rain gauge sensors were tested for the existence of significant differences between them. The most advanced
algorithm was the Spatial Consistency Check (SCC) which identified outliers by comparing observed values with
data from neighbouring stations.
An important outcome of the system was the determination of the quality index ($QI$) of analysed data, which
is a unitless value with a range [0.0, 1.0], where "0.0" means extremally bad data and "1.0" means perfect data.
This $QI$ metric was determined by the RainGaugeQC for each sensor and then the sensor with higher quality is
taken for further processing.

**Table 2. A summary of the quality control algorithms used in the RainGaugeQC system before and after**
**modification.**

| Abbr. | Algorithm | Before modification | After modification |
|---|---|---|---|
| GEC | Gross Error Check | Gross errors | |
| RC | Range Check | Exceeding climatological thresholds | |
| RCC | Radar Conformity Check | Detection of false rainy and non-rainy events | |
| BSC | Blocked Sensor Check | Detection of blocked sensors | |
| TCC | Temporal Consistency Check | Comparison of two sensors | Time series comparison with weather radar data |
| BC | Bias Check | – | Detection and correction of bias with adjusted radar data |





| SCC | Spatial Consistency Check | Detection of outliers from the local vicinity | Detection of outliers from the local vicinity (updated) |
|---|---|---|---|


## 3.2 Directions of development in RainGaugeQC

The possibility of incorporating non-professional data at IMGW became a motivation for more sophisticated data
quality control. The QC algorithms in the previous version of RainGaugeQC proved unsuitable for non-
professional data, as they are often subject to greater uncertainty than from professional rain gauges, and besides,
these gauges are generally not dual-sensor. On the other hand, the inclusion of new data significantly improved
the performance of the SCC algorithm due to the higher density of the measurement network. Therefore, it was
necessary to redesign the RainGaugeQC system in order to adapt it to rain gauge networks equipped with different
types of sensors, supervised to various degrees, so that the system became more universal. The modified algorithms
tailored to the new challenges associated with incorporating non-professional data are summarised in Table 2 in
the "After modification" column.
*TCC*. In the new version of the TCC (time series comparison with adjusted weather radar data) algorithm,
weather radar data is used to compare time series from a specific time interval to check the correlation between
rain gauge measurements and radar observations. The correlation coefficient is used as a metric for the relevant
component of the quality index of the rain gauge data. This allows a reduction in the data quality index of rain
gauges with measurements disturbed for a certain time period due to failure, poor maintenance or bad location.
*BC*. The above TCC algorithm is not sensitive to the bias of rain gauge measurements, so the BC (bias check
with adjusted radar data) algorithm is used to detect bias in the data. It also works by analysing long-term data
series, but in this case they are used to compare data accumulations from rain gauges with radar accumulations.
The quantitative estimation of the bias of the rain gauge data allows relevant components of the quality index to
be determined. In the case of private rain gauges, unbiasing is carried out as well as reducing the $QI$ value.
*SCC*. The SCC (detection of outliers from the local vicinity) algorithm was already introduced in the first
version of the RainGaugeQC system, but significant modifications have been made to the current version. It detects
outliers, i.e. the measurements at a given time-step deviate from the values from rain gauges located in a certain
area. The increase in the number of rain gauges through incorporating non-professional data has made it easier to
determine the degree of outlying for individual data. The quality index reduction for outliers is quantified on the
basis of the spatial variability of the precipitation field derived from the radar data.

## 3.3 New version of TCC algorithm (Time series comparison with weather radar data)

The TCC algorithm is designed to eliminate erroneous rain gauge measurements ($G$) by analysing the correlation
on long time series. The reference is radar precipitation ($R$) after adjustment with rain gauge observations only
from professional networks.
For the calculation, pairs of rain gauge ($G$) and radar ($R$) data are taken if at least one of the values is greater
than 0.025 mm, and their quality index ($QI$) is at least 0.7 for $G$ and 0.8 for $R$. Two time series aggregated from
10-min accumulations: "long" and "short" comprising 10 and 5 days, respectively, are analysed. For long series
the number of non-precipitation pairs $c_{dry}$ is determined provided that both values are less than 0.025 mm. For





each series hourly accumulations are determined and then the number of measurement pairs c and correlation
coefficient r are calculated.
The procedure for assessing data quality is carried out by checking a list of conditions. For a given measurement
these conditions are examined sequentially and, depending on the result, further ones are checked or the quality
index is reduced accordingly.
The check is stopped if the accumulations of both radar and rain gauge precipitation for the long series are
below the assumed threshold values:
$\qquad \left(\sum_{10\ \text{days}}(R) < 3.0\right)$ and $\left(\sum_{10\ \text{days}}(G) < 6.0\right) \rightarrow$ TCC stopped $\qquad$ (1)
If the amount of radar precipitation for the long series is below the assumed threshold and the amount of rain
gauge precipitation is above the corresponding threshold, then the check is also stopped, but the quality of the rain
gauge data is reduced by a value of 0.05:
$\qquad \left(\sum_{10\ days}(R) < 3.0\right)$ and $\left(\sum_{10\ days}(G) \geq 6.0\right) \rightarrow$ TCC stopped, $QI = QI - 0.05$ $\qquad$ (2)
This indicates that there are large differences between the two accumulations, but because the rainfall recorded
by the radar is too low, the calculation of the correlation coefficient may be not reliable in such cases.
The check is passed if the number of measurement pairs is above the preset threshold and correlation coefficient
is above 0.3 for short or long series. Then the quality index is reduced on the basis of the relevant correlation
coefficient, according to the following formula:
$\qquad (c > 6)$ and $(r > 0.3) \rightarrow$ TCC passed, $QI = \begin{cases} QI & r > 0.85 \\ QI - \frac{1-r}{4} & r \leq 0.85 \end{cases}$ $\qquad$ (3)
If there is an insufficient number of measurements for short series and at the same time the number of non-
precipitation data pairs is above a preset threshold, indicating that there is a longer non-precipitation period, then
the TCC is stopped and $QI$ is reduced:
$\qquad \left(c_{dry} > 1000\right)$ and $\left(c_{short} \leq 6\right) \rightarrow$ TCC stopped, $QI = QI - 0.05$ $\qquad$ (4)
Finally, the number of measurements and correlation coefficient with radar data for short and long periods are
examined. If the condition in Formula 5 is met then the check is stopped. If not, the check is failed:
$\qquad [(c \leq 6)$ or $(r = "no\ data")] \rightarrow$ TCC stopped, $QI = QI - 0.05$ $\qquad$ (5)
$\qquad$ else $\rightarrow$ TCC failed, $QI = QI - 0.3$.
This formula applies to cases when there are too few measurements, or the correlation coefficient could not be
calculated or was below the assumed threshold for short or long series.
**3.4 New algorithm BC (Detection of bias with adjusted radar data)**
The determination of bias in the BC algorithm is carried out by comparing the precipitation accumulations obtained
from the time series recorded on a given rain gauge with adjusted radar rainfall as a reference. For the most recent
10 days using a 10-min temporal resolution, rain gauge and radar precipitation accumulations, denoted as $\Sigma G$ and
$\Sigma R$ respectively, are calculated from gauge-radar pairs, for which both measurements have a quality index of at
least 0.7 for $G$ and 0.8 for $R$.





Choice of the length of the precipitation accumulation period to determine the bias is not a trivial issue. Long
accumulations better reflect the overall uncertainty of the measurements at a given station, but, on the other hand,
short accumulations better follow the current precipitation characteristics during a particular precipitation event.
Most often, bias is determined on rainfall accumulations from up to a few dozen hours, but sometimes on much
longer accumulations – e.g. Yousefi et al. (2023) used seasonal totals to unbias radar data with rain gauge data.
The $bias$ of the rain gauge measurements is calculated from the ratio of radar to rain gauge precipitation
accumulations:
$bias = \frac{\Sigma R}{\Sigma G}$                   (6)
The bias determined in this way is used to reduce the quality index $QI$ of the controlled rain gauge data. If the
precipitation accumulations $\Sigma G$ and $\Sigma R$ are similar, which is checked using the corresponding similarity function,
the quality of the measurement remains unchanged. The similarity function is defined as follows:
$1.3 \cdot \min(\Sigma G, \Sigma R) + 7.0 > \max(\Sigma G, \Sigma R)$        (7)
If the radar and rain gauge precipitation accumulations for a given rain gauge are not similar, then depending
on the bias determined from Formula 6, the value of the quality index $QI$ of a given measurement is reduced, but
to a varying extent, according to the formula:
$QI = \begin{cases} QI - 0.05 & bias \in \left[\frac{1}{5}, 5\right] \\ QI - 0.2 & bias \in \left[\frac{1}{10}, \frac{1}{5}\right) \text{ or } bias \in (5, 10] \\ QI - 0.5 & bias \in \left[\frac{1}{20}, \frac{1}{10}\right) \text{ or } bias \in (10, 20] \\ QI - 1.0 & bias \in \left(0, \frac{1}{20}\right) \text{ or } bias \in (20, +\infty) \end{cases}$   (8)
In cases when the bias cannot be estimated, the $QI$ of a particular measurement is reduced according to the
formula:
$QI = \begin{cases} QI - \min\left(1.0, \frac{|\Sigma G - \Sigma R|}{10.0}\right) & (\Sigma G = 0.0) \text{ or } (\Sigma R = 0.0) \\ QI - 0.2 & (\Sigma G = \text{"no data"}) \text{ and } (\Sigma R = \text{"no data"}) \end{cases}$   (9)
In terms of data from private weather stations, they are considered subject to much greater uncertainty due to
the lack of supervision of the technical condition of the rain gauges, poor maintenance, bad location, etc. Such
stations should therefore be treated more rigorously than stations supervised by the institutions responsible for the
measurements. The similarity function (Formula 7) is not applied, as their $QI$ values are always reduced by the
formula:
$QI = \begin{cases} QI - 0.1 & bias \in \left[\frac{1}{5}, 5\right] \\ QI - 0.3 & bias \in \left[\frac{1}{10}, \frac{1}{5}\right) \text{ or } bias \in (5, 10] \\ QI - 0.7 & bias \in \left[\frac{1}{20}, \frac{1}{10}\right) \text{ or } bias \in (10, 20] \\ QI - 1.0 & bias \in \left(0, \frac{1}{20}\right) \text{ or } bias \in (20, +\infty) \end{cases}$   (10)
when $bias$ cannot be estimated, the $QI$ value of a given measurement is reduced by the formula:



$$QI = \begin{cases} QI - \min\left(1.0, \frac{|\Sigma G - \Sigma R|}{10.0}\right) & (\Sigma G = 0.0) \text{ or } (\Sigma R = 0.0) \\ QI - 0.4 & (\Sigma G = \text{"no data"}) \text{ and } (\Sigma R = \text{"no data"}) \end{cases} \qquad (11)$$

In addition, unbiasing should be performed for data from private stations, which is not done for other types of
stations, as they only have a reduced $QI$. Unbiasing is performed on the basis of the bias determined from Formula
6, but limiting its value to factor 4:
$$bias_4 = \begin{cases} \frac{1}{4} & bias \leq \frac{1}{4} \\ bias & \frac{1}{4} < bias \leq 4 \\ 4 & bias > 4 \end{cases} \qquad (12)$$

The above limitation on the value of the $bias_4$ factor is to protect against too large a change in the value of the
corrected precipitation (van Andel, 2021).
Finally, the unbiased precipitation accumulation $G_{cor}$ is determined from the formula:
$$G_{cor} = bias_4 \cdot G \qquad (13)$$

As IMGW does not yet have a sufficiently dense network of cooperating private stations (Droździoł and
Absalon, 2023), tests have not been carried out to verify the algorithm designed in this study on data from such a
network.
**3.5 Updated SCC algorithm (Detection of outliers from the local vicinity)**
The spatial methods for quality control, such as the SCC, are especially effective for dense rain gauge networks
because they utilise observations from nearby stations (Alerskans et al., 2022). Thus, when applied to sparse
networks, it is more likely that a correct value measured by a rain gauge will be classified as erroneous in the case
of intense convective rainfall of a very local nature.
Based on the analysis of the performance of the SCC algorithm – as published in a previous paper on the
standard version of RainGaugeQC system (Ośródka et al., 2022) in Appendix C – a modification was made in
relation to the degree of $QI$ reduction depending on the spatial variability of rainfall.
The algorithm has not changed in terms of assigning each rain gauge measurement to one of the three classes
of outliers: strong, medium, and weak, and additionally non-outlier. However, the algorithm for reducing the $QI$
value of each measurement assigned to any of the outlier classes was modified. In the current version of the
algorithm, the magnitude of $QI$ reduction depends on whether a given rain gauge measurement is within an area
of a high spatial variability of precipitation determined from weather radar data of sufficient quality $QI(R)$. In this
case, the outlier is treated less restrictively. The concept of spatial variability function ($SVF$) was introduced for
this purpose, and is defined as follows:
$$SVF = \frac{SVF_{mean}(R_{mean}) + SVF_{var}(R_{var})}{2} \qquad (14)$$

The $SVF$ consists of two components indicating the degree of spatial variability of the precipitation:
$$SVF_{mean}(R_{mean}) = \begin{cases} 1 & R_{mean} \geq 1.0 \text{ mm} \\ \frac{R_{mean} - 0.1 \text{ mm}}{1.0 \text{ mm} - 0.1 \text{ mm}} & 0.1 \text{ mm} < R_{mean} < 1.0 \text{ mm} \\ 0 & R_{mean} \leq 0.1 \text{ mm} \end{cases} \qquad (15)$$



$$SVF_{var}(R_{var}) = \begin{cases} 1 & R_{var} \geq 1.0 \text{ mm}^2 \\ \frac{R_{var} - 0.03 \text{ mm}^2}{1.0 \text{ mm}^2 - 0.03 \text{ mm}^2} & 0.03 \text{ mm}^2 < R_{var} < 1.0 \text{ mm}^2 \\ 0 & R_{var} \leq 0.03 \text{ mm}^2 \end{cases},$$

where $R_{mean}$ is the mean radar precipitation (in mm) for wet pixels in the 100 km x 100 km subdomain including 25 km margins (see: Ośródka et al., 2022); $R_{var}$ is the mean variance of radar precipitation (in mm2) in the subdomain calculated analogously to $R_{mean}$.

On the basis of the value of the $SVF$ function, the reduction in the quality index for individual rain gauge observation is determined, according to its classification into a specific outlier class:

$$QI = \begin{cases} QI - (0.30 \cdot (1 - SVF) + 0.10 \cdot SVF) & \text{strong outlier} \\ QI - (0.20 \cdot (1 - SVF) + 0.05 \cdot SVF) & \text{medium outlier} \\ QI - 0.10 \cdot (1 - SVF) & \text{weak outlier} \end{cases} \qquad (16)$$

### 3.6 Determination of $QI$

Before all the checks, each rain gauge observation is assigned the perfect $QI$ value (1.0). Depending on the result of a particular QC algorithm, the $QI$ of an examined measurement is decreased by a relevant value. If the final $QI$ value, i.e. after all checks, is below a preset threshold, the observation is considered useless and is replaced with "no data".

## 4 Analysis of the RainGaugeQC system performance on non-professional data

The performance of the RainGaugeQC system, designed to control the quality of precipitation data from professional and non-professional rain gauge networks, is shown through a comparison of the statistics calculated for these two rain gauge networks:

- professional network of IMGW, the Polish NMHS, supplemented in the border region by data from CHMU, which is the Czech NHMS,
- non-professional network of the General Directorate of the State Forests.

The most important characteristics of these networks are summarised in Table 1, and the locations of the rain gauges are shown in Fig. 1. Rain gauges from private networks have not been included, as the establishment of their network at IMGW is still at a preliminary stage.

The analysis was carried out for four months – April, July, October 2023 and January 2024 – considered typical of the four seasons. The summer season (July) is dominated by convective precipitation, which is often intense and highly variable in time and space, while the winter season (January) is dominated by stratiform precipitation, often in the form of snow. In the intermediate seasons (April, October) precipitation is less intense – it is generally rain, and is rarely convective.

### 4.1 Verification metrics

The reliability of the precipitation estimates generated using the RainGaugeQC system was verified by comparison with the reference precipitation accumulations from manual rain gauges that are treated as the closest to the true precipitation at their locations. The following metrics were employed:

- Pearson correlation coefficient:





$$CC = \frac{\sum_{i=1}^{n}(E_i - \overline{E})(O_i - \overline{O})}{\sqrt{\sum_{i=1}^{n}(O_i - \overline{O})^2 \sum_{i=1}^{n}(E_i - \overline{E})^2}}$$ (17)

- root mean square error:

$$RMSE = \sqrt{\frac{1}{n}\sum_{i=1}^{n}(E_i - O_i)^2}$$ (18)

- root relative square error:

$$RRSE = \frac{\sqrt{\sum_{i=1}^{n}(E_i - O_i)^2}}{\sqrt{\sum_{i=1}^{n}(O_i - \overline{O})^2}}$$ (19)

- statistical bias:

$$BIAS = \frac{1}{n}\sum_{i=1}^{n}(E_i - O_i)$$ (20)

where $E_i$ is the estimated value, $O_i$ is the reference value, $i$ is the gauge number, $n$ is the number of gauges, whereas $\overline{E}$ and $\overline{O}$ are the mean values of $E_i$ and $O_i$, respectively.

**4.2 Non-professional versus professional rain gauge data**

A comparison of reliability metrics of precipitation estimates obtained from a network of professional and non-professional rain gauges is shown in Fig. 4. Point measurements of rainfall were verified against values at rain gauge locations obtained from the interpolation of manual rain gauges using the inverse distance weighting method. Professional rain gauges situated at manual gauge locations, a relatively common situation in the IMGW network, were not included in the statistics in order not to favour this category of data. Therefore, around 200 professional rain gauges were used for verification instead of all 469.

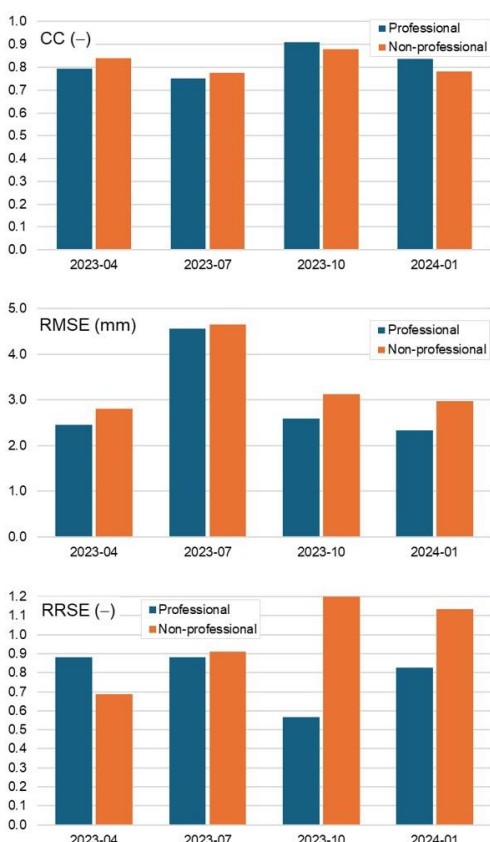

**Figure 4: Reliability statistics of rainfall estimates calculated for data obtained from the network of professional (navy) and non-professional (orange) rain gauges. Spatially interpolated manual rain gauges are used as a reference. Data from April, July, October 2023 and January 2024.**

The reliability of the non-professional data in general is close to that of the professional data, especially as regards the correlation coefficient: on average for both it is about 0.82, and the differences between them are small, at below 0.06. The RMSE metric related to the deviation from the reference data is already clearly worse for the non-professional data, by on average about 0.41 mm. The largest difference was found for January, when it reached 0.65 mm. Only in the summer period (July) is the difference between the non-professional and professional data small (0.09 mm), though the error values are highest at that time (4.65 and 4.55 mm, respectively). During this period, convective precipitation is frequent, more intense, and also more dynamic, and as a consequence, the comparison with spatially interpolated reference data can produce large differences. In contrast, a similar but relative RRSE metric gives less conclusive results: in April it is much better for the non-professional data (0.69 versus 0.88), while in the other months the non-professional data are worse than the professional, with a significant difference of 0.63 in October.



### 4.3 Comparison of the QC system performance on professional and non-professional data

In this Section an examination is made of the extent to which the $QI$ of rain gauge data for professional and non-professional stations is reduced by the RainGaugeQC system in different months of the year. The $QI$ plays a key role in the multi-source precipitation field estimation performed by the RainGRS system as the $QI$ index is one of the most important weights during spatial interpolation of rain gauge data and, most importantly, it is a weight when rain gauge data is combined with the other precipitation estimates – radar and satellite-based. As a result of this approach, the impact of low-quality data on the final precipitation field estimate can be reduced.

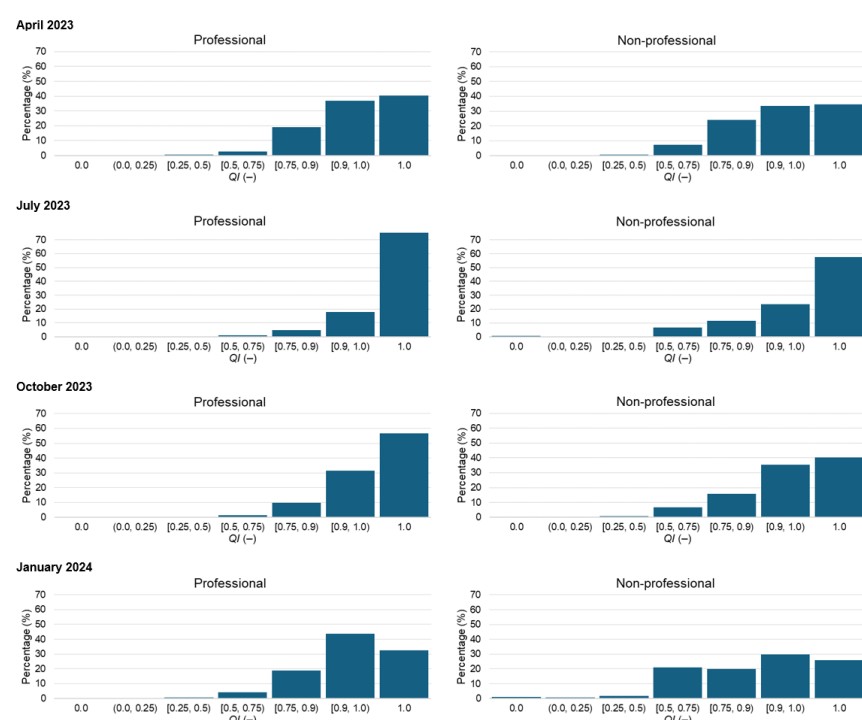

**Figure 5: Percentages of data with $QI$ values in different ranges (histograms). Data from April, July, October 2023 and January 2024.**

Fig. 5 summarises the percentage of rain gauge data in different ranges of $QI$ values assigned to individual measurements as a result of $QI$ performed with a modified version of the RainGaugeQC system for four months representing different seasons, separately for professional and non-professional stations. It can be noted that, in general, $QI$ values are significantly higher for professional data, meaning that QC algorithms indicate higher uncertainty in non-professional data. While unreduced quality ($QI = 1.0$) characterises 32.5 – 76.1% of all professional data depending on the season, just 26.0 – 57.6% of non-professional data. On the other hand, lower values below $QI < 0.75$ at different seasons characterise 1.4 – 4.9% of the professional data and 7.4 – 24.2% of the non-professional data.





There is noticeable seasonal dependence of the number of data with *QI* in specific value ranges, which is similar
for professional as well as non-professional data. The highest percentage of data with a *QI* of exactly 1.0, i.e.
perfect data according to the RainGaugeQC system, is observed in July (summer) and equals 76.1% and 57.6%
for professional and non-professional data respectively, while the percentage of data with poor qualities is also
lowest in this month for both types of the data: 1.4% and 7.2%, respectively. Considering the distribution of *QI*
values in the different ranges, the data from January proved to be the least reliable, when the percentage of data
with low *QI* values, i.e. in the range between 0.0 and 0.75, is the highest, reaching 4.9% for professional and 24.2%
non-professional data.
**4.4 Impact of non-professional rain data on the reliability of precipitation estimates**
The following data sets were applied to test the influence of non-professional rain data on the reliability of
precipitation estimation: (i) professional only and (ii) professional and non-professional together after quality
control with the modified version of RainGaugeQC. From both rain gauge data sets, 10-min multi-source estimates
of precipitation accumulations were generated with the RainGRS system and then aggregated to the daily
accumulations. Table 3 shows the reliability metrics of the daily accumulations calculated for April, July, October
2023 and January 2024, using the manual rain gauge data as a reference. Statistics were determined at the locations
of the manual rain gauges.

**Table 3. Reliability metrics of estimates of daily RainGRS precipitation accumulations generated using rain**
**gauge data: professional and professional with attached data from non-professional rain gauges after**
**quality control with the modified version of RainGaugeQC. Measurements from manual rain gauges are**
**used as a reference, data from April, July, October 2023 and January 2024.**

| Rain gauge networks | CC (–) | RMSE (mm) | RRSE (–) | BIAS (mm) |
|---|---|---|---|---|
| *April 2023* | | | | |
| Professional (IMGW and CHMU) | 0.832 | 2.74 | 0.64 | 1.36 |
| Professional (IMGW and CHMU) and non-professional (State Forests) | 0.872 | 2.40 | 0.55 | 1.11 |
| *July 2023* | | | | |
| Professional (IMGW and CHMU) | 0.835 | 3.99 | 0.57 | 1.03 |
| Professional (IMGW and CHMU) and non-professional (State Forests) | 0.847 | 3.71 | 0.55 | 0.93 |
| *October 2023* | | | | |
| Professional (IMGW and CHMU) | 0.920 | 2.35 | 0.43 | 0.91 |
| Professional (IMGW and CHMU) and non-professional (State Forests) | 0.922 | 2.28 | 0.41 | 0.79 |
| *January 2024* | | | | |
| Professional (IMGW and CHMU) | 0.844 | 2.55 | 0.65 | 1.42 |
| Professional (IMGW and CHMU) and non-professional (State Forests) | 0.846 | 2.52 | 0.64 | 1.40 |






It can be seen from Table 3 that after the incorporation of non-professional data provided by the General
Directorate of the State Forests into RainGRS, all reliability metrics improved. The correlation coefficient, CC,
increased for all months analysed only marginally. Greater improvement after the inclusion of non-professional
data can be seen in all metrics related to error magnitude: RMSE, RRSE and BIAS, which on average decreased
by 0.13 mm, 0.02, and 0.08 mm, respectively.
Analysing the four metrics used, the most positive impact of incorporating non-professional data was found in
April 2023, an intermediate month, when all characteristics improved: CC increased by 0.04, while metrics related
to error magnitude decreased: RMSE by 0.34 mm, RRSE by 0.09 and BIAS by 0.35 mm. This observation is
consistent with the results shown in Fig. 4, where in April the non-professional data were even more reliable than
the professional data in terms of CC and RRSE metrics. The smallest impact of non-professional data was observed
in January, when the improvement was negligible.
It should be pointed out that the number of non-professional rain gauges available for this study was not large:
the ratio between the number of rain gauges in the non-professional and professional networks was about 1:4.
Therefore, it can be expected that if there were more of these non-professional rain gauges, then the benefit from
them in terms of improvement in the reliability of the precipitation estimates would be even more pronounced.
This impact is not only due to the measurement information provided by these rain gauges, but also largely due to
the fact that additional rain gauges make quality control of all rain gauges much more effective.
**4.5 Impact of non-professional rain gauges on estimated multi-source precipitation field – varying impact**
**in different locations**
This section presents two case studies illustrating the influence of non-professional precipitation data on the
reliability of precipitation estimates generated by the RainGRS system. The location of the study areas is shown
on a map of Poland (Fig. 6). Locations in central Poland were chosen because the network of professional rain
gauges is sparsest there (see Fig. 1), so the influence of non-professional data on the final estimate of the
precipitation field can be expected to be more evident. Two different RainGRS precipitation field estimates were
generated using rain gauge data: (i) from professional rain gauges only, (ii) from both professional and non-
professional rain gauges. The impact of incorporating non-professional rain gauge data on multi-source field
estimates was assessed using manual rain gauge measurements as reference data. The analyses were conducted on
daily accumulations because only this kind of data are available from manual rain gauges.



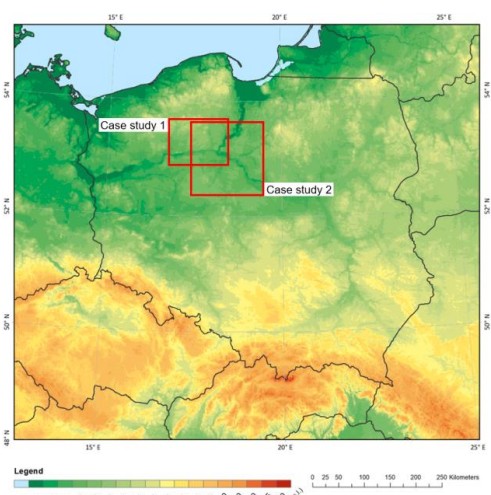


**Figure 6. Location of case studies on a map of Poland.**


### 4.5.1 Case study 1: isolated convective precipitation (29-30 July 2023)

On 29 and 30 July 2023 Poland was under the influence of a trough of low pressure and atmospheric front systems moving from west to east. There were some showers and thunderstorms with precipitation locally reaching more than 60 mm per day, which triggered flash flooding in major cities in the north of the country. Fig. 7 presents the daily precipitation accumulations for this day, which shows the effect of including non-professional rain gauge data to the input data to the RainGRS model generating multi-source precipitation field estimates.

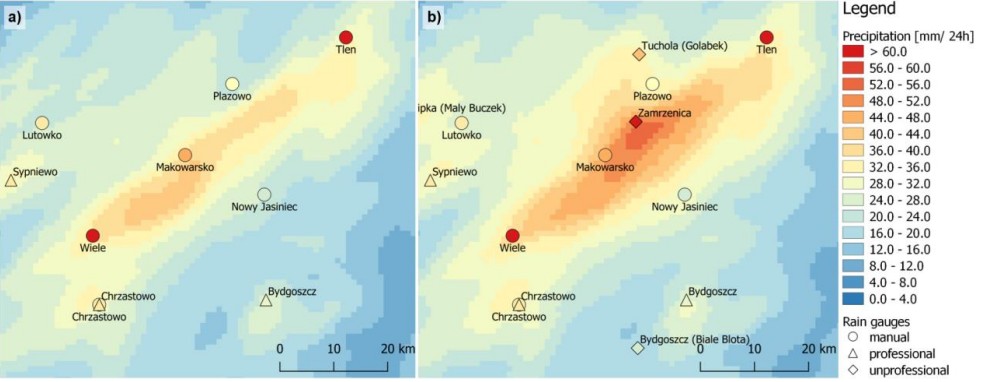

**Figure 7: Precipitation maps of multi-source RainGRS estimates from: a) professional, b) professional and non-professional data. The symbols are filled with colours that correspond to the precipitation values measured by each rain gauge. A fragment of Poland, daily accumulations from 29.07.2023, 06 UTC to 30.07.2023, 06 UTC.**


In the fields of estimated precipitation accumulations in the vicinity of the thunderstorm cell in Fig. 7, it can be seen that after incorporation of the non-professional data, the accumulations became noticeably higher, as the





data from the non-professional rain gauges are generally higher than those from the professional ones – a general
increase in values can be seen in Fig. 7b compared to Fig.7a. Using the measurements from the manual rain gauges
as reference data, it can be concluded that the obtained increase in the estimated RainGRS precipitation field is
closer to the reference precipitation (this is confirmed by the results in Table 3). Regarding the thunderstorm cell
moving through the study area, it was compact, small in size (its diameter was about 10 km) and no professional
rain gauge was in its path. It was detected by weather radars, so it is visible on the multi-source estimate, but the
precipitation values are underestimated compared to the reference precipitation recorded by the manual rain gauges
located in the path of this cell.
When including the non-professional data, a rain gauge in Zamrzenica on the route of this storm cell measured
a daily rainfall of 62.3 mm, resulting in a significant increase in the RainGRS precipitation estimate in this area:
from 31.6 to 50.6 mm at the Zamrzenica location. However, due to the small number of rain gauges in the area,
the high precipitation spread over a much larger region than the close vicinity of the cell. This is evidenced by the
lower precipitation measured by the manual rain gauge at Nowy Jasiniec (23.3 mm), while the precipitation
estimate increased here from 24.2 to 31.0 mm.
Closest to the path of the cell was the Makowarsko manual rain gauge, which measured 46.8 mm. The multi-
source estimate after including the non-professional rain gauge increased from 37.8 to 47.1 mm, which is in very
good agreement with the reference value. The precipitation estimate at the Płazowo manual rain gauge location
also increased: from 22.4 to 33.5 mm, while this rain gauge measured 29.2 mm. The increase in estimates was
therefore too high, but nevertheless, after data from non-professional rain gauges were added to the estimate, it
was closer to the measurement from the reference rain gauge. The highest value of 68.5 was measured by the Tleń
manual rain gauge, but the incorporation of the non-professional data only slightly improved the highly
underestimated estimate from 31.5 to 33.7 mm.
**4.5.2 Case study 2: winter stratiform precipitation (3-4 January 2024)**
At the beginning of January 2024, Poland was in the range of low-pressure systems moving from west to east and
associated atmospheric fronts. Rainfall and sleet were observed, with snowfall in the north-east of the country and
in the mountains in the south. In the north and centre, there was also freezing rain causing glaze. The example
shown in Fig. 8 relates to a lowland area in central Poland, like in the first case study, but here there was stratiform
precipitation, which was significantly lower but at a greater extent, as is typical for winter.

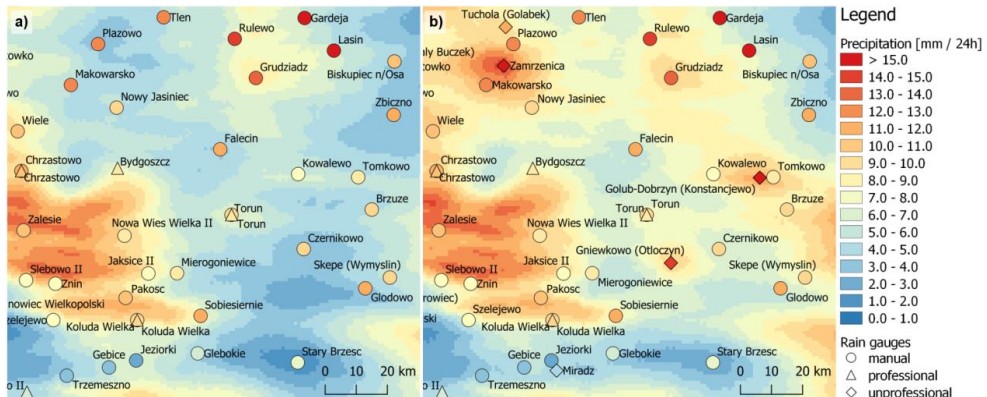

**Figure 8: Precipitation maps of multi-source RainGRS estimates from: a) professional, b) professional and non-professional data. The symbols are filled with colours that correspond to the precipitation values measured by each rain gauge. A fragment of Poland, 24-h accumulations from 3.01.2024, 06 UTC to 4.01.2024, 06 UTC.**

The RainGRS precipitation field estimation generated values that were underestimated compared to the manual rain gauge measurements: the estimated values were lower by 3.2 mm on average, while their daily accumulation averaged 9.5 mm at the locations of these rain gauges. This is mainly due to the underestimation of weather radar and, to a lesser extent, telemetric measurements.

The inclusion of data from non-professional rain gauges, despite their small number, increased the RainGRS estimate at manual rain gauge locations by an average of 1.3 mm. For example, it can be seen that that Zamrzenica non-professional rain gauge had a positive effect on the estimated daily precipitation accumulation (RainGRS) at the manual rain gauge located in Płazowo, where 12.1 mm was measured, and the estimates with and without the incorporation of non-professional data were 9.5 and 3.3 mm, respectively.

The impact of the Miradz non-professional rain gauge was slightly different. It measured a value of 3.1 mm and caused the estimates at the location of the two closest manual rain gauges to decrease at Jeziorki from 6.7 to 4.7 mm, and at Gębice from 6.0 to 4.9 mm, approaching the values from the manual rain gauges of 1.7 and 3.0 mm respectively. On the other hand, the influence of Miradz appeared to negatively affect the estimates at the manual rain gauge locations of Kołuda Wielka and Szelejewo, where values that had been underestimated compared to the reference rainfall were lowered even further.

The analysis of the two case studies shows that data from non-professional rain gauges, despite their generally somewhat greater uncertainty, can in most cases play a positive role in the estimation of the precipitation field.

## 5 Conclusions

Data from non-professional rain gauge networks, as additional source of precipitation data, increase the density of available rain gauge networks. In consequence they can improve precipitation field estimates at high spatial resolution and can be very helpful to NHMSs for various meteorological and hydrological applications. However, advanced data quality control systems are required to make these data useful for operational applications. At the same time, it should be possible to objectively quantify the uncertainty associated with each individual measurement.





The RainGaugeQC system, applied to quality control of rain gauge data, was redesigned in order to adapt it to

different rain gauge networks supervised to various degrees. In a modified version of the TCC algorithm, more

sophisticated data control was developed applying weather radar data, taking into account various aspects of data

quality, such as consistency analysis of data time series. The new BC algorithm was introduced to detect bias of

rain gauge measurements comparing rain gauge and radar long-term accumulations. In the SCC algorithm,

significant modifications have been made to quantify the quality index reduction for outliers on the basis of the

spatial variability of the precipitation field derived from the radar data. The performance of the modified system

was verified based on independent measurement data from manual rain gauges, which are considered one of the

most accurate measurement instruments. The influence of incorporating non-professional precipitation data on

reliability of multi-source precipitation estimates generated by the RainGRS system was also analysed.

The main conclusions derived from the analyses carried out in this study can be summarised as follows:

1.   The incorporation of data from non-professional stations into professional rain gauge data, even if they

are of poorer quality (Fig. 5), nevertheless improves the reliability of the estimated multi-source

precipitation field (Table 3), but on the condition that advanced quality control is carried out.

2.   Despite the quality control performed, the influence of individual rain gauges on the precipitation field

estimates may sometimes not be positive, as can be seen from the examples shown in Section 4.5.

Furthermore, the same rain gauge may have a different influence, positive or negative, on an estimated

precipitation field in various places.

3.   Precipitation field estimates provided by weather radar data play a very important role in the developed

RainGaugeQC algorithms. However, it is necessary to perform their advanced quality control beforehand

and to adjust them with rain gauge measurements.

4.   An important benefit of including data from non-professional networks is the improvement in

performance of individual QC algorithms. This is especially true for the spatial consistency check (SCC),

in which the density of a rain gauge network is crucial.

5.   IMGW is in the process of setting up a network of private rain gauges. After its operational launch, it will

become possible to test the QC algorithms proposed in this paper on data from these rain gauges.

**Acknowledgement**
The work described in this paper was carried out primarily within the COSMO consortium (Consortium for Small-
scale Modelling) as Priority Task EPOCS "Evaluate Personal Weather Station and Opportunistic Sensor Data
Crowdsourcing" during the period 2023-2024.

*Code availability*. The data processing codes are protected through the economic property rights to the software
and are not available for distribution. The codes used for processing follow the methodologies and equations
described herein.

*Data availability*. Out of the data used in this article, the following are publicly available:
IMGW rain gauge data in the form of 10-minute accumulations: https://danepubliczne.imgw.pl/pl/datastore, tabs:
„Dane archiwalne" / „Dane meteorologiczne" / „*year*" / „Meteo_*year-month*.zip" / „B00608S_*year_month*.csv"
(B00608S is the code for the 10-min rainfall parameter).





Radar data as 1-h files of precipitation accumulation (PAC) maps: https://danepubliczne.imgw.pl/pl/datastore,
tabs: „Dane archiwalne" / „Mapa zbiorcza sumy opadów za godzinę." / „*year*" / „*month*" /
COMPO_PAC.comp.pac_*year-month-day*.tar.
Other data used in this article is available upon request, provided it is not restricted by its producer.
*Author contributions*. KO, JS, and AJ designed algorithms of the RainGaugeQC system. KO developed the
software code and performed the simulations. JS, KO, AJ, and AK prepared the paper. JS made figures. AK carried
out statistical calculations.
*Competing interests*. The contact author has declared that none of the authors has any competing interests.

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
