# Peer review of "Adaptation of RainGaugeQC algorithms for quality control of rain gauge data from professional and non-professional measurement networks"

_Atmospheric Measurement Techniques, 2024_

## Author Comment (AC1)

**Summary**

The article describes a number of performant quality control functions in order to assess the quality of precipitation measured by ground based stations. The overall system is detailed and provides a good understanding of different observation problems. The final outcome of the RainGaugeQC system is a quality index for each station.

**General remarks**

lines 67 and 73: There are numerous stations which are operated by former meteorologists or even active meteorologists on their private ground which provide data of a high quality. Often, operators of such stations are organised in amateur meteorological clubs. These clubs often publish data on the internet and togther with the conditions of the measurement locations, so that they are documented. Therefore, private stations do not guarantee a good data quality, but a considerable number of them is regularly monitored and their data of high quality. This needs to be verified for each private station, though.

We are aware of this. However, in general, we cannot say anything with certainty about the fulfilment of WMO standards by private stations. Besides, the reliability of individual stations for very different reasons can change over time. Nevertheless, we have somewhat mitigated the meaning of this passage (line 75).

line 82: internationally, dual-sensor rain gauges are not the rule, even in national weather services. The WMO classifies station location (e.g. GIMO Guide of WMO (2021)) and has performed several gauge intercomparison exercises (Lanza / Vuerich, 2009).

We are also aware that two-sensor stations are not standard. We have removed the sentence from lines 82-83, especially as it does not apply to non-professional stations.

**Specific remarks**

Please note: all line numbers below refer to the original version of the article, not modified.

line 33: "... highly distorted" - depending on the operator, usually radar measurements are quantitatively less accurate, but not highly distorted (see e.g. WMO, 2024)

This is a debatable issue. We have been working for many years on weather radar data quality control (RADVOL-QC system, references in the article) and in our opinion the radar data is "highly distorted". This is not visible on the data that comes to the users. However, we have removed the word "highly".

line 33: "Rain gauge measurements are still considered ..." - maybe you could be more specific, like: "In hydrology, rain gauge measurements are considered ..."

**Thanks - we have changed it.**

line 42: it is correct that the authors have published relevant papers in this context. However, the reader would also appreciate more general publications, such as WMO BPG on radar data quality and Lanza / Vuerich (2009) on the WMO rain gauge intercomparison.

**Thanks for this reference - we have added it.**

lines 110 and 115: radar data should only be incorporated after their quality control - else this is not state of the art (WMO, 2024).

Thanks also for this reference - we were not familiar with this document! We have added a relevant note to the text in line 110:

"but only after quality control (WMO-No. 1257, 2024)"

line 117: If you do not set a minimum threshold, your correlation risks to give you random results - this should be pointed out more clearly here.

We have added information on the threshold assumed:

"so a minimum precipitation threshold should be used to filter out data from such periods"

line 131: the adjustment to rain gauges modifies the radar time series temporally and thus bears the risk that a comparison to a rain gauge time series becomes more difficult. A time series with corrected radar data and without gauge adjustment might give better results, in particular if the gauge data to be analysed have been integrated into the adjustment.

**We have modified the text:**

"However, such measurements are not common, so remote sensing data such as radar observations, which are more widely available, can be used as a benchmark, but they require the QC to have been previously carried out."

lines 132 and 133: please give an indication of the length of the required time interval, e.g. one year.

de Voss (2019) used a 14-day interval (for 5-min data). We have added this information in line 133.

table 1: the column "type of rain gauges" would merit additional information: what is the minimum volume of the tipping bucket gauges? Are they unheated for the two first rows? Which measurement type is the third row? "heated" does not tell much ...

**Minimum volume of the tipping bucket gauges applied at IMGW is 0.1 mm.**

line 181: please comment on the accuracy of the daily gauges if there is rainfall at the time of the day change. How accurate can the daily precipitation amount be under such conditions? Manually operated gauges often have a time interval for the operator to read and check the rainfall amount which may be in the order of 15 minutes. Such information should be communicated, additionally to the formulation that their data "are believed to be more accurate".

This observation is confirmed by studies carried out on data from the IMGW rain gauge networks, e.g. Urban and Strug (2021). We have added this paper to the references and the text at line 182:

"..., which has been confirmed by extensive reliability analyses of different types of rain gauges at IMGW (Urban and Strug, 2021)."

line 189: Please comment on the range of 250 km. Please note that for hydrological quantitative use, distances beyond 120 km range are subject to higher uncertainties in the radar measurement due to the measurement height and measurement geometry. Which range is practically used for your applications?

Radars of POLRAD network measure reflectivity at a range of 250 km as standard, but these measurements are applied for different purposes. We are aware that for QPE these 250 km are

far too far. On the other hand, all radar networks in Europe measure at ranges greater than 120 km (as far as we know, the minimum is 180 km).

In our case, we use data up to 215 km. This distance is the result of a compromise: (i) as short a range as possible, (ii) full coverage of the whole country with measurements. We didn't describe this in the article because radar measurements only play a supporting role in it. But since you find it useful, we have added the following in line 189:

"For the estimation of the precipitation field, data up to 215 km from the radar are used. This distance represents a balance between achieving the shortest possible range and ensuring complete coverage of the entire country with measurements."

Figure 3: Why 215 km ranges? Can you please elaborate on this?

The explanation is as above.

line 210: please give more details on the satellite data used in the system! Which is the data source, how is it quantitatively transformed to precipitation? Which is the original resolution in time and space and how is it mapped to fit to the ground measured data?

We have not written more extensively about satellite precipitation because it is a side issue for this paper, although it is of course of interest. However, there is a reference in our paper to Jurczyk et al. (2020b), where this is described in detail.

Table 2: What do you understand with "gross errors"? Please explain more in detail or refer to the correspondent section in this paper

We have added sentences in the text in line 228:

"The GEC involves detecting when natural limits are exceeded, while the RC focuses on identifying when climate-based thresholds are surpassed at an individual gauge."

Table 2: TCC - over which time interval does the comparison take place?

Details of the particular algorithms, including parameter values, are given in the relevant chapters. Time intervals in the TCC are given in line 276.

Table 2: SCC - please provide more details on the definition of outliers in this context!

We write about the SCC in chap. 3.5. However, there only the changes with respect to the original version (see Ośródka et al., 2022) are described. We have not repeated this basic information about the algorithms in the present work.

line 255: what would be a typical "specific time interval"? Please provide a range in minutes!

In Section 3.2 there is only a introduction to the directions in which the changes in RainGaugeQC have gone. All details are given in the relevant subsections of Section 3. The time interval in the TCC is 5 and 10 days - we write about this in line 276.

line 256: do you request a minimum amount of rainfall in radar and gauge?

Yes. We write about this in line 275 (0.025 mm).

line 256: When do you consider a correlation coefficient to be "good"? Please give more details in the assessment of the correlation quality!

We write about the relationship between the correlation coefficient and the assigned QI value in chapter 3.3 (formulae 3-5).

line 263: how do you carry out the unbiasing procedure? Please provide more details!

The BC (unbiasing) procedure is described in detail in section 3.4.

line 272: when do you consider a time series to be long? What is the minimum duration for this?

This is described in detail in section 3.3.

line 275: the amount of 0.025 mm is per which time period?

This 0.025-mm threshold is applied to 10-min totals. We write about this in line 276.

Regarding lines 255-275: After reading some of the Reviewer's previous comments on lines 255-275, we concluded that at the beginning of Section 3.2 of the article we did not clearly emphasise that here is only a preview of what has been changed, while the details will be developed in Sections 3.3 - 3.5. We have added the relevant information to the text in line 253:

"Here is a brief overview of the changes made to the RainGaugeQC algorithms, whereas detailed information can be found in Sections 3.3 to 3.5."

line 336: the formula implies that the QI value is reduced even for perfect data. Is this intended?

We agree: only for private stations we always reduce the quality by at least 0.1, which is due to their generally lower reliability. We have reason to assume that while professional stations meet WMO standards and are properly maintained and supervised, we are not so sure about private stations.

Formulas (9) and (11): it is unusual to work with a bias in this way - more often, a multiplicative approach is used (see chapter 3.3.5 of WMO, 2024). Your approach penalises a deviation of 5 mm equally for a rain gauge sum of 10 mm and of 50 mm where in the first case, this represents 50%, in the second one 10%.

Bias is applied multiplicatively (equation 14). Regarding the example of the 5 mm difference in rainfall of 10 and 50 mm: according to the similarity function (equation 7), these cases are similar to each other. Thus, if this example concerns professional/unprofessional rain gauges, then the QI is not reduced in both cases. However, if it applies to private rain gauges, then the bias is 1.5 and 1.1 respectively, i.e. they were in the range [0.2, 5.0] (equation 10), so in both cases QI = QI - 0.1.

We use the reduction of QI due to Bias with caution due to some uncertainty also in the radar rainfall R (especially for the 10-min accumulation period), and even after adjustment it is not ground truth.

Figure 4: (a) please indicate the number of dry time spells for each of the months - for some statistics this plays an important role

To avoid the negative impact of dry time spells, we used a threshold of 0.5 mm for daily totals - days with lower rainfall were not included. The number of such days per month was as follows: Apr - 4, Jul - 7, Oct - 5, Jan - 5. We have added this information to line 415:

"Days with precipitation accumulation below 0.5 mm were not included in the calculations (in total 21 days in these four months)."

(b) how did you take into account the systematic bias of extreme values due to the interpolation of the gauges? Does the distance of each gauge play a role? How would the result have looked when comparing to radar data?

We are aware that interpolation of rain gauge data during convective precipitation can involve large errors depending on the distance from the nearest rain gauge. Therefore, professional rain gauges situated at manual gauge locations, which is a relatively common situation in the IMGW network, were not included in the statistics in order not to favour this category of data (lines 413-414). The average distance of professional and unprofessional rain gauges from manual ones was then similar.

line 422: how can you state that the reliability of both data sources is comparable, if your reference is biased? This is, also in the light of the important and illustrative discussion in this paragraph, a statement without foundation.

We have tried to minimise the bias associated with the interpolation of rain gauge data as we have written in response to the previous comment.

line 458: Your finding that gauge data in Junauary are the least reliable is at least surprising since point rainfall data in summer are less representative in space. It would therefore be beneficial to read a discussion on these findings, in particular considering the predominant rainfall types and their spatial variability. Is this influenced by low temperatures in winter?

We have added such an explanation to the text, line 460:

"The low percentage of QI = 1.0 in January for both data types is due to the methodology used to determine these values in the SCC algorithm (Section 3.5). It uses the spatial variability function (SVF), which quantifies the spatial variability of precipitation at each time step. The high variability of precipitation is associated with convective precipitation and the introduction of the SVF function is intended to prevent such precipitation from being treated too rigorously and decreasing QI of good measurements. However, convective precipitation is very rare in winter in Poland, hence the frequent reduction of QI for weak outliers."

line 521: is this a finding for this day only or are the non-professional gauges always biased towards higher values?

Table 3 shows that this is a general feature of non-professional gauges.

line 540: do you consider this high value to be an outlier or a true value? If a true value, please discuss the discrepancies that you can see in figure 7.

Fig. 7 shows a convective cell with a very small area and very high rainfall intensity. Precipitation from such cells is generally short-lived. The reasons for these discrepancies are probably: (1) the lack of professional rain gauges in the path of this cell, which resulted in a decrease of the multi-source estimate, as well as ineffective adjustment of radar data to the rain gauges, which also effected the multi-source estimate, (2) the radar measures instantaneous values of rainfall, so with a 5-min time step it may have missed the maximum rainfall.

We write about this in lines 524-528.

line 556: do you have an explanation for the underestimation? Was there snowfall?

This is explained in a modified sentence in lines 557-558:

"This is mainly due to the underestimation of weather radar, which has problems detecting precipitation from low clouds most commonly occurring in winter, usually as snow, at farther distances from the radar site."

**Technical details**

line 11: replace "10-min" by "10 min".

line 49: delete "often"

lines 76, 133: replace "np" by "e.g."

line 79: delete "very"

line 122: I suggest to replace "underestimated" by "underestimating the true rainfall"

line 141: replace "were" by "are"

lines 147 - 148: please provide the number of stations here or omit them from the two following bullet points in order to give a uniform information

line 292: please add "r" after "correlation coefficient" for clarity purposes!

We have included all of these above technical remarks in the article. Thank you for reviewing it so carefully!

**References:**

WMO (2024) Guide to Operational Weather Radar Best Practices (WMO-No. 1257). Volume VI: Weather Radar Data Processing. Provisional version at https://community.wmo.int/en/activity-areas/weather-radar-observations/best-practices-guidance

Lanza, L., Vuerich, E. (2009) The WMO Field Intercomparison of Rain Intensity Gauges. Atmospheric Research, Volume 94, Issue 4, December 2009, Pages 534-543.

WMO (2021) Guide to Instruments and Methods of Observation (WMO No. 8). https://community.wmo.int/en/activity-areas/imop/wmo-no 8.

Thank you for these references!

---

## Author Comment (AC2)

The authors modify an existing rain gauge quality control (QC) algorithm, to make it better able to handle data from non-professional gauges. The modified algorithm is used to quantify the accuracy of both professional and non-professional data. The added value of including the non-professional data in multi-source precipitation products is demonstrated, both for a longer period and for individual events.

There is a growing interest in using non-professional precipitation observations to complement professional observations from national or regional meteorological and hydrological services. A major challenge concerns the higher degree of uncertainties and errors in the non-professional data. Therefore, efforts to develop and share practically applicable QC algorithms are timely and must be encouraged. I find the approach presented overall sensible (although some further clarification is needed; see below). The presentation is clear and the calculations and analyses appear well performed, as far as I can judge, but some revision is needed in light of the following comments.

**General:**

Much of the results and conclusions are based on measurements from four single months, considered typical for their corresponding season. This is a quite heavy assumption, firstly because one month is very short in a rainfall/precipitation context – it may contain just a few single events – and secondly because a single month may differ substantially from seasonal climatology. I would suggest to include as much "seasonal data" as you have available for this analysis. Alternatively, but much less preferably, show that the selected months are sufficiently good "seasonal representatives" for supporting the conclusions made.

We understand the reviewer's approach, which suggests to perform verification on data reflecting the climatology of precipitation. However, in this case this is, on the one hand, impossible due to the heterogeneity of the data on which such an analysis could be carried out and, on the other hand, the purpose of our analysis was different:

- 1. The radar network in Poland was completely replaced between 2021 and early 2023, and moreover two new radars were installed. Therefore, it is not possible to carry out verification on data from these years, and on the other hand, the radars operating earlier (installed in 2002-2004) were of a different type, with single polarisation and therefore quality control was less advanced. Since radars play a key role in the current version of RainGaugeQC, verifying its performance on data from old radars would give a distorted results of its performance.
- 2. During the same period most of the telemetric rain gauges were upgraded (from tipping-bucket to weighing ones) and their number significantly increased, so the rain gauge data also changed completely in terms of their number and quality, which drastically affects the effectiveness of the individual RainGaugeQC algorithms.
- 3. We initially thought of verifying on the whole four seasons (Dec-Feb, etc.), but found that more important are the characteristic types of precipitation in Poland: snow in winter and deep convection in summer. Our intention was to investigate how the developed algorithms perform with different types of precipitation, and so we chose the four months in which these types of precipitation occur most frequently. Currently, snowfall in Poland does not last all winter, it is most frequent in January (in January 2024: 12 days in Warsaw located in the centre of Poland), and the most intense storm precipitation occur in July. Stratiform rainfall is most common in October, and weak convective rainfall starts to appear in April.

So we did not aim to carry out analyses that took into account the climatology of precipitation. This, of course, would also be interesting, but would require reasonably homogeneous rain gauge and also radar data series.

On lines 347-349 is written that (sufficiently dense) PWS data is not available at IMGW, and therefor tests have not been made on PWS data. I wonder whether the EUMETNET Sandbox with Netatmo data (Netatmo, 2021), which cover Poland, could be used for this purpose. If so, this would be a very interesting addition to the paper.

We downloaded Netatmo data, which for Poland has good coverage. However, there were several reasons why we did not use them:

- We only had access to data from 2020 but these data were not suitable because this is the period before radar replacement. The newer data are not available for free.
- The 2020 data are, at least for Poland, collected in an unclear way: often time steps between successive measurements are irregular (from several to several tens of minutes) without any information on accumulation time. This uncertainty would introduce too much error in the statistics.

We are still waiting for the network of PWS stations at IMGW, which is currently being set up, but we would have to wait next two years for the data (sufficient station numbers plus time to collect measurements).

Parts of the description of the updated algorithm are a bit hard to follow (see examples below and also comments from RC1), more explanation and justification needed.

Thanks to the comments of both Reviewers, we hope that we have sufficiently improved the transparency of the algorithm descriptions.

We have added a paragraph on line 283:

"First, the 10-day radar precipitation total  $\sum_{10 \text{ days}}(R)$  is checked. If this is too low, then the correlation coefficient is not calculated, as it may not be reliable in such a case. In addition, it is checked whether the rain gauge rainfall  $\sum_{10 \text{ days}}(G)$  differs significantly from the radar data (formulae 1 and 2) and depending on this, the quality index of G is reduced."

We changed the sentence in lines 283-284:

"If the both accumulations are below the assumed threshold values, then the quality index of the rain gauge data is not reduced and the check is stopped:"

We also changed the sentence in lines 286-288:

"If the amount of radar precipitation for the long series is below the assumed threshold and the amount of rain gauge precipitation is above the corresponding threshold, indicating large differences between the two accumulations, then the check is also stopped and the quality index of the rain gauge data is reduced by 0.05:"

We have removed the sentence in lines 290-291.

The English is overall understandable and not a big problem for me, but there are examples of curious expressions that could be improved by a "native check", I think (some examples below).

We have improved the English in the article. We hope that the text is now noticeably better.

**Specific:**

• 68: I think the P in PWS usually stands for Personal (although I do have seen also Private).

We have corrected to "personal"

• 76 and others: Change np. to e.g.

We have corrected.

• 79: Example of sub-optimal English (in my opinion): "relatively very large".

We have corrected.

• 124: et al. is missing.

We have added.

• Table 1: What type is the DLP gauge? Weighing?

We have added.

• 171: This expression is rather for the Introduction.

In the Introduction there is already information about the dependence of measurement uncertainty on device type. We have therefore left it at line 171.

• Figs. 2 and 3: I suggest combine into one.

The figures with the distribution of manual rain gauges and radars are in different Sections (2.1 and 2.2), so combining them would be unclear to readers. We will leave the decision to the editor of the journal.

• 204-207: Some more details here are needed to understand the following applications.

We have added in line 207:

"This adjustment is carried out from gauge-radar ratios determined at rain gauge locations, spatially interpolated over the entire domain."

• 238: Is this done at each time step? Do you mean the sensor with highest quality?

Yes, for each step - we have added this information in line 238: "This *QI* metric..."  $\rightarrow$  ,,At each time-step this *QI* metric..."

Yes, with the highest QI. In the case of two sensors, "higher" could also be used in our opinion, but to avoid confusion we have changed to "the highest".

• 246: "proved unsuitable", in what way?

The sentence in lines 246-248:

"The QC algorithms in the previous version of RainGaugeQC proved unsuitable for nonprofessional data, as they are often subject to greater uncertainty than from professional rain gauges, and besides, these gauges are generally not dual-sensor."

we have shortened into: "The QC algorithms in the previous version of RainGaugeQC turned out to be inadequte for non-professional data, as these gauges are generally not dual-sensor.", to highlight the key reason.

• 255: Which time step (or interval)?

There is only a very general description of the ideas for the various algorithms here, while their detailed descriptions can be found in later subsections of this chapter. We have added the sentence in line 253:

"Here is a brief overview of the changes made to the RainGaugeQC algorithms, whereas detailed information can be found in Sections 3.3 to 3.5."

• 268: English: "degree of outlying"

We have corrected:

"to determine the degree of outlying for individual data"  $\rightarrow$  "to determine the degree to which individual data is an outlier"

• 276: How was 5 and 10 days selected? They are not that different, can you justify that they are short and long enough?

In general, the idea is to test the correlation on a data series that is as short as possible, but still representative. The 5-day sequences optimally take into account the current correlation of the data. But we have found that, for example, during periods of low rainfall, the correlation calculated on short-time series can be random and then a 10-day sequence can give a more reliable correlation.

We have removed the word "long" from line 272.

We changed a sentence in lines 275-276:

"Two time series aggregated from 10-min accumulations: "long" and "short" comprising 10 and 5 days, respectively, are analysed.

 $\rightarrow$  "Two time series aggregated from 10-min accumulations: "short" and "long" comprising 5 and 10 days, respectively, are analysed in order to test correlations on time series that are as short as possible and, on the other hand, sufficiently representative."

• Section 3.3: Generally, many numbers/thresholds here that are given without explanation or motivation. I assume they have been carefully set, but some clarification would be good.

We have added a new paragraph after line 269:

"All parameters of the algorithms described in sections 3.3 to 3.5 were selected empirically by comparing the calculated QI values with the expected ones derived from our assessment of the data reliability."

• 285: Why different limits for R and G?

Regarding equation (1): if the precipitation is low and the correlation may not be reliable, then if the radar confirms this low precipitation, we do not calculate the correlation and decrease the QI. However, we allow some tolerance, hence the different thresholds for R and G.

In formula (2): if rain gauge precipitation (G) is high and radar precipitation (R) is low, but this difference must be evident, then we decrease the QI but do not calculate the correlation as it would not be reliable. Hence, the threshold on G is significantly higher than on R.

We have revised the text relating to formulae (1) and (2) to improve clarity. Details of these changes are in response to an earlier Reviewer's comment, which we have marked with the symbol (\*).

• 295: Has c in this eq. been defined?

Yes, "c" is defined in line 278 (we have corrected the missing italic in this symbol).

• 323: This eq. is one example of something that needs more clarification.

We have changed the notation of equation (7) to make it clearer:

 $SF(\Sigma G, \Sigma R) = \begin{cases} \text{true} & 1.3 \cdot \min(\Sigma G, \Sigma R) + 7.0 > \max(\Sigma G, \Sigma R) \\ \text{false} & 1.3 \cdot \min(\Sigma G, \Sigma R) + 7.0 \le \max(\Sigma G, \Sigma R) \end{cases}$

• 336: So, PWS is always biased? Is this a reasonable assumption?

The reduction of the quality by at least 0.1 is due to their generally lower reliability. We have reason to assume that while professional stations meet WMO standards and are properly maintained and supervised, we are not so sure about private stations.

• 359: English: "additionally non-outlier"

We have changed "and additionally non-outlier" into: "or to the class of correct data".

• 373: How is this classification made? More explanation needed here.

We have completed the sentence in line 373, before the colon: "(see: Ośródka t al., 2022)"

• 411: Insert ", respectively," between "gauges" and "is".

We have corrected.

• Fig. 5: The font is a bit small.

We have increased the size of all the fonts in this figure.

• 458: Any influence of snow?

Probably yes. We have added this observation in line 460:

"Probably the reason for the worse results for January is the occurrence of snowfall, which is more challenging for radars to detect."

• 570-571: A bit strong statement, in my opinion, that these two cases show that non-professional are useful "in most cases". Probably/hopefully they are, but the statement would require more cases to be evaluated.

We changed in line 571:

"can in most cases play a positive role in the estimation of the precipitation field."  $\rightarrow$  "is likely to play a positive role in the estimation of the precipitation field in many cases."

• 597-599: Is this a conclusion from the present study? I do not really see this.

The Reviewer is right. We have removed this point from the Conclusions.

• 602-604: This is clearly not a conclusion from this work. I suggest extend this to a final paragraph about future efforts, remaining issues, etc.

Thank you for this suggestion! We have moved point 5 of the conclusions into a separate paragraph and completed it:

"The development of the quality control system for telemetric rain gauge measurements will be continued. Plans include incorporating precipitation data from other non-professional networks to supplement the IMGW rain gauge network. This will increase the proportion of data with potentially lower reliability, which may require even more sophisticated algorithms for the quality control. Moreover, IMGW is in the process of establishing a network of personal rain gauges. Once this network is operational, it will be possible to test the quality control algorithms proposed in this paper using data from these rain gauges."

**Reference:**

Netatmo (2021): EUMETNET Sandbox: Netatmo observing network data v1. NERC EDS Centre for Environmental Data Analysis, 2025-03-15. https://catalogue.ceda.ac.uk/uuid/e8793d74a651426692faa100e3b2acd3/

We thank the Reviewer for this information. We used Netatmo data from 2020 in our initial work on RainGaugeQC, but we do not have access to this data from the period for which we were able to test the current version of RainGaugeQC.

---

## Author Response (AR2)

Dear Editor,

Thank you for your decision and we send back our article in its final form. Following your suggestion, we have combined Figs. 2 and 3 into one Fig. 2 with two panels. However, we have decided not to move the marked case study areas from Fig. 6 (now Fig. 5) to either of the panels in the current Fig. 2, as these figures are very far from each other as they are on pages 6 and 19. We felt that it would be inconvenient for the reader to go back to check the locations of these two areas.

Yours sincerely,
Jan Szturc